# Prevalence of Bovine Viral Diarrhea Virus Infections in Pigs on Jeju Island, South Korea, from 2009–2019 and Experimental Infection of Pigs with BVDV Strains Isolated from Cattle

**DOI:** 10.3390/vetsci9030146

**Published:** 2022-03-21

**Authors:** SeEun Choe, Seong-in Lim, Gyu-Nam Park, Sok Song, Jihye Shin, Ki-Sun Kim, Bang-Hun Hyun, Jae-Hoon Kim, Dong-Jun An

**Affiliations:** 1Virus Disease Division, Animal and Plant Quarantine Agency, Gimchen 39660, Gyeongbuk-do, Korea; ivvi59@korea.kr (S.C.); saint78@korea.kr (S.-i.L.); changep0418@korea.kr (G.-N.P.); ssoboro@korea.kr (S.S.); shinji227@korea.kr (J.S.); kisunkim@korea.kr (K.-S.K.); hyunbh@korea.kr (B.-H.H.); 2Veterinary Medicine College, Jeju University, Jeju-si 63243, Jeju-do, Korea; kimjhoon@jejunu.ac.kr

**Keywords:** CSFV, immune response, BVDV-1, BVDV-2, pig

## Abstract

On Jeju Island, South Korea, pigs have not been vaccinated against classical swine fever (CSF) since 1999. Analysis of bovine viral diarrhea virus (BVDV) isolated from pigs on Jeju Island between 2009 and 2019 identified five BVDV-1a strains and one BVDV-1b strain. These BVDV types were shown to be the same types as BVDV strains isolated from neighboring cow farms. BVDV antibody-positive pigs (both BVDV-1 and -2) were also detected at 54 of 168 pig farms during this period. In pig infection experiments using BVDV-1a and -2a strains isolated from neighboring cow farms, BVDV-1a was detected in the blood of one of four pigs infected at both 6 and 35 days post-infection (dpi) and in the blood of two of the four pigs at 28 dpi. Pigs showed higher anti-BVDV-1 titers (5.5 ± 1.5 log_2_) at 35 dpi. BVDV-2a was detected in the blood of one of four pigs infected with this virus at 28 dpi only, and lower antibody titers (2.75 ± 0.75 log_2_) were seen in these pigs at 35 dpi. While BVDV infection is not particularly pathogenic in pigs, it is still important to monitor porcine BVDV infections due to a differential diagnosis of CSFV.

## 1. Introduction

Bovine viral diarrhea virus (BVDV) is a single-strand, positive-sense RNA virus belonging to the genus *Pestivirus* within the family *Flaviviridae* [1]. The genus *Pestivirus* includes animal pathogens that are of worldwide socioeconomic significance; these include BVDV (*Pestivirus A–B*), classical swine fever virus (CSFV, *Pestivirus C*), and border disease virus (BDV, *Pestivirus D*) [1]. Other *Pestiviruses* include *Pestivirus E* (pronghorn pestivirus), *Pestivirus F* (Bungowannah virus), *Pestivirus G* (giraffe Pestivirus), *Pestivirus H* (Hobi-like pestivirus), *Pestivirus I* (Aydin-like pestivirus), and *Pestivirus J* (rat pestivirus) [1]. Bovine viral diarrhea (BVD) is an important disease as it causes great economic loss to cow farmers worldwide [2]. BVDV has two genotypes, type 1 and type 2, which are classified into sub-genotypes: BVDV-1 (1a to 1u; 21 sub-genotypes) and BVDV-2 (2a to 2d; 4 sub-genotypes) [3]. Infection of pigs with BVDV usually occurs without clinical signs, allowing the virus to spread without detection. However, several previous studies suggested that BVDV causes anemia, rough skin, growth retardation, atrophy, and diarrhea in piglets, in addition to reproductive disorders and recurrent abortion in pregnant sows [4,5,6]. BVDV infection in pigs was first reported in Austria in 1954 and subsequently reported in Germany, the Netherlands, China, and the UK [2,7,8,9,10]. In pigs, BVDV infection is caused by mixed-breeding livestock using BVDV-contaminated vaccines, feeding of cattle-derived material to pigs, and wild rodent BVDV carriers [5,11]. The antigenic cross-reactivity between BVDV and CSFV led to a diagnostic error when CSF occurred in the Netherlands in 1997 [12]. The differential, serological diagnosis of these viruses is very important for the detection of CSF antibodies in nonvaccinated regions and in (normally) CSF-free areas [13,14]. On Jeju Island, which is located off the southernmost tip of mainland South Korea, CSF vaccination has not been implemented since 1999. However, frequent detection of CSF-antibody-positive pigs was confirmed as being due to contamination of the live attenuated CSF vaccine strain [15]. However, some CSF antibody-positive cases are thought to be due to infection by BVDV, although this has not been reported formally.

The purpose of this study was to investigate the prevalence and cause of BVDV infection in pigs in the Jeju Island region from 2009 to 2019, and to provide information about BVDV infection via clinical observations and immunological and pathological analyses of BVDV-1a and -2a infection patterns in experimentally infected pigs.

## 2. Materials and Methods

### 2.1. Virus Isolation from Samples

CSF antibody and antigen detection is conducted at least twice a year on all Jeju Island pig farms (about 300 farms). Since Jeju Island is a non-CSF vaccine region, it is essential to perform different diagnosis with BVD antibody when CSF antibody and antigen are detected. Between 2009 and 2019, CSF antibodies and antigens were detected on 168 pig farms. To identity the prevalence of BVDV on CSF-positive pig farms, 734 CSF antibody-positive blood samples were tested for the presence of BVDV antigens and antibodies. A total of 60 cow fecal samples were also collected from cow farms in the vicinity of pig farms with BVDV-infected pigs (five samples per cow farm; *n* = 12 farms). Madin-Darby bovine kidney (MDBK; ATCC CCL-22) cells were used to isolate BVDV from the blood of pigs and the feces of cows. MDBK cells were cultured in 6-well plates in alpha-MEM containing 5% (*v*/*v*) horse serum (Gibco-BRL, New York, NY, USA) and 1X antibiotics and antimycotic solution (Gibco-BRL, New York, NY, USA) in a 5% (*v*/*v*) CO_2_ incubator at 37 °C. Cow fecal samples (filtered using 0.2 µm filters) and pig plasma suspensions were added to MDBK cells. After 1 h, the suspension fluid was removed and the infected cells containing 1X antibiotics and antimycotic solution were cultured in a 5% (*v*/*v*) CO_2_ incubator at 37 °C. After culturing for 72 h, BVDV was measured using an immunofluorescence staining assay (IFA). BVDV isolated from samples performed 4–5 passages by MDBK cells, and harvested by freezing and thawing infected cells three times, followed by centrifugation at 1500 rpm for 10 min to remove lysed cells. The virus content of cell lysates was determined using the Reed–Muench method.

### 2.2. RT-PCR for Antigen Detection

To detect BVDV, total RNA was extracted from blood and fecal samples using a microcolumn-based QIAamp Viral RNA Mini kit (Qiagen, Maryland, MD, USA). The specific primers used to amplify the partial 5′ untranslated region (5′UTR) gene were 85F (5′-GCG AAG GCC GAA AAG AGG CTA-3′) and 391R (5′-TCC ATG TGC CAT GTA CAG CAG AGA-3′). The cDNA was amplified synthesized at 50 °C for 50 min using a one-step RT-PCR kit (Inclone com, Seongnam, Korea). The PCR conditions were as follows: 95 °C for 15 min, followed by 40 cycles of 95 °C for 20 s, 55 °C for 40 s, 72 °C for 40 s, and then a final 72 °C incubation for 5 min. PCR products of the expected size (245 bp) were cloned using the pGEM-T Vector System II™ (Promega, Wisconsin, WI, USA) and sequenced at the Macrogen Institute (Macrogen Co. Ltd., Seoul, Korea) using T7 and SP6 sequencing primers and an ABI Prism^®^3730_XI_ DNA Sequencer (Lifetechnologies com, California, CA, USA).

### 2.3. Phylogenetic Tree Analysis

The partial 5′UTR nucleotide sequences of 37 reference BVDVs from around the world were obtained from GenBank and assigned to two groups: group 1 (subgroups 1a–1q) and 2 (subgroups 2a–2b). For phylogenetic analysis of 9 Korean BVDV isolates and the 37 reference strains from GenBank, multiple sequence alignment (including the reference sequences) was performed using CLUSTAL X (ver. 2.1.) [16]. Phylogenetic trees, based on the nucleotide sequences of the 5′UTR genes, were constructed using the maximum-likelihood (ML) method in the MEGA 7.0 software (https://www.megasoftware.net, accessed on 1 January 2022) [17], with nucleotide distances (*p*-distance) and 1000 replications used for bootstrap analysis. Prior to each phylogenetic analysis, the model of nucleotide substitution that best fitted the data was identified using the find-best model (Tamura-Nei model) in MEGA 7.0.

### 2.4. Neutralizing Antibody Assay

Two-fold serial dilutions of pig serum samples were mixed with equal volumes of either a BVDV type 1a strain (08GB44-1, Accession no: JQ418633) or a BVDV type 2a strain (08Q723, Accession no: JQ418635) at 37 °C for 1 h. These samples were then used to infect MDBK cells, and the ability of neutralization antibodies in the sera to prevent entry of BVDV into MDBK cells was measured by IFA. In brief, after culturing for 72 h in a 5% (*v*/*v*) CO_2_ incubator at 37 °C, cells were incubated with a specific WB210 anti-BVDV-1 monoclonal antibody (APHA com., cat. no. RAE0823, Addlestone, UK) or a specific anti-BVDV-2 monoclonal antibody (VMRD com., cat. no. BA-2, Pullman, Washington, WA, USA) for 1 h. Binding of the primary antibodies was detected with antimouse FITC antibody (KPL com, Massachusetts, MA, USA), which was visualized using immunofluorescence microscopy Nikon Eclipse Ti (Nikon Instruments, Melville, NY, USA).

### 2.5. Experimental Infection with BVDV-1 and -2 Strains

Healthy 70-day-old Jeju pigs, all free from porcine circovirus 2, porcine reproductive respective syndrome virus, CSFV, and BVDV antigens and antibodies were used in this study. Animal experiments were performed at 25 °C and at 60–70% humidity. A total of 12 pigs were divided into three groups (four pigs per group), which were inoculated with BVDV-1a (JLI-1a-c strain; group 1), BVDV-2a (SSH-2a-c strain; group 2), or no virus (group 3). The BVDV-1a and BVDV-2a strains used were isolated from cows. Pigs inoculated with BVDV strains were given 2 mL of virus (titer, 10^5.5^ TCID_50_/_mL_) both intranasally and intramuscularly. Blood, feces, and nasal swab samples were collected from each pig at 0, 3, 6, 10, 14, 21, 28, and 35 days postinoculation (dpi), and BVDV was detected by RT-PCR. The collected blood samples were used to determine SN titers (0, 6, 14, 21, 28, and 35 dpi) and leucocyte cell counts (0, 3, 6, 10, and 14 dpi). All pigs were euthanized at 35 dpi and their organs (tonsil, lung, heart, spleen, liver, kidney, mesentery lymph node, and lymph node) were analyzed to detect BVDV antigen.

## 3. Results

### 3.1. Antigen and Antibody against BVDV from Pigs in Jeju Island

Of 734 CSF antibody-positive pig samples, 171 harbored anti-BVDV antibodies (*n* = 165) or antigens (*n* = 6). Between 2009 and 2012, BVDV was detected in pigs from six pig farms on Jeju Island (a CSF vaccine-free region); however, no virus was detected in any pigs between 2013 and 2019. The strain designations of the viruses (based upon the designation of pig farms from which they were isolated) are shown in Figure 1A. Three strains of BVDV in 2010 were also isolated from cows cohabiting in two pig farms: a BVDV-1a type strain (JLI-1a-c) and a 1b type strain (JLI-1b-c), both from a JLI farm; and a BVDV-2a type strain (SSH-2a-c) from an SSH farm (Figure 1A). The five BVDV-1a strains isolated from Jeju Island pigs (JYK-1a-p, KMS-1a-p, SSH-1a-p, JLI-1a-p, and GSK-1a-p strains), and the JLI-1a-c cow strain, have high nucleotide sequence similarity (98.6–99.5%) to the NADL strain (data not shown). The pig KTG-1b-p strain and the cow JLI-1b-c strain have high nucleotide sequence similarity (98.1%) to the CP7-5A strain (data not shown). Of the 168 pig farms that were CSF antibody- and antigen-positive, 54 also harbored BVDV; the number of BVDV antibody-positive cases ranged from one to six. Anti-BVDV antibodies were detected in pigs from 24 farms in 2009, and from between one and seven pig farms in each year from 2010 to 2019 (Figure 1B). From 2009 to 2019, anti-BVDV antibodies were detected in pigs at 54 pig farms: 46 of these had anti-BVDV-1 antibodies and 8 had anti-BVDV-2 antibodies (Figure 1B). The mean antibody range was determined as 5–8 log_2_ for anti-BVDV-1a and 4–6 log_2_ for anti-BVDV-2a, respectively (data not shown).

### 3.2. Phylogenetic Tree for BVDV

The 5′UTR gene sequences of the 9 Korean BVDV strains, isolated in this study, and 37 reference BVDV when a maximum-likelihood (ML) tree was constructed using the Tamura-Nei model (Figure 2). The nine Korean BVDV strains consisted of six strains (SSH-1a-p, JYK-1a-p, KMS-1a-p, JLI-1a-p, GSK-1a-p, and KTG-1b-p) derived from pigs and three strains (JLI-1a-c, JLI-1b-c, and SSH-2a-p) derived from cows (Figure 2). On the ML tree, five strains (SSH-1a-p, JYK-1a-p, KMS-1a-p, JLI-1a-p, and GSK-1a-p) derived from pigs and one strain (JLI-1a-c) from a cow aligned with the BVDV-1a subgroup, which contains strain NADL (accession no. AJ133738; Figure 2). The pig KTG-1b-p strain and the cow JLI-1b-c strain aligned with the BVDV 1b subgroup, which contains strain CP7-5A (accession no. AF220247), whereas the cow SSH-2a-c strain aligned to the BVDV 2a subgroup, which contains the strain C413 (accession no. AF002227; Figure 2).

### 3.3. BVDV and Anti-BVDV Antibody Detection in Experimentally Infected Pigs

Leukopenia was not observed in pigs in any of the three infection groups (BVDV-1a, -2a, or mock) and white blood cell counts remained within the normal range (13,000–36,000/uL) during the observation period (0–14 dpi; Figure 3A).

Pigs inoculated with BVDV-1a showed average antibody titers against BVDV-1a as follows: 1.75 (log_2_) at 21 dpi, 4.5 (log_2_) at 28 dpi, and 5.5 (log_2_) at 35 dpi (Figure 3B). Pigs inoculated with BVDV-2a showed somewhat later seroconversion than pigs inoculated with BVDV-1a, with titers of 2 (log_2_) at 28 dpi and 2.75 (log_2_) at 35 dpi (Figure 3B). RT-PCR of group 1 detected BVDV-1a in the blood of one pig at 6 dpi, two pigs at 28 dpi, and one pig at 35 dpi; however, BVDV-2a was detected in the blood of only one pig from group 2 (at 28 dpi) (Table 1). However, neither BVDV-1a nor BVDV -2a were detected in any fecal or nasal samples (Table 1).

### 3.4. BVDV Antigen Detection in Pig Tissues

All pigs were euthanized at 35 dpi and their organs harvested for analysis. No virus was detected in any of the organs recovered from pigs inoculated with the BVDV-1a strain (Table 2). By contrast, virus was detected in most of the organs analyzed from at least one of the pigs inoculated with BVDV-2a: lung (*n* = 1), spleen (*n* = 1), liver (*n* = 1), kidney (*n* = 2), mesenteric lymph nodes (*n* = 1), inguinal lymph nodes (*n* = 1), and ileum (*n* = 1) (Table 2). However, the histopathological analysis of tissues confirmed to be BVDV-2-positive by RT-PCR showed no abnormal lesions nor any of the pathological features associated with CSFV infection.

## 4. Discussion

The prevalence of BVDV antibody-positive pigs was reported to be 6.8% in Denmark, 3–40% in Australia and Germany, 3.2% in Ireland, 2.2% in Norway, and 3.0% in Canada [2,7,8,9,10,11,18,19]. BVDV infection of pigs in China was confirmed to be one strain (BVDV-1 strain ZM-95) [20], and the rate of BVDV seroconversion in pigs in Shanghai in 2007 and 2008 was 35.9% and 64.1%, respectively [21]. BVDV infection in pigs occurs via direct or indirect contact with ruminants [8], by experimental infection [22], and by exposure to contaminated vaccines. Enhanced infection rates have been reported in animals coinfected with BVDV and other viral or bacterial pathogens [23,24]. On Jeju Island, there is a strong correlation between the emergence of BVDV-positive pig farms and their close proximity to either cow pastures or pig and cow breeding facilities. Thus, the cause of the BVDV infections in pigs on Jeju Island is likely to be contamination directly or indirectly from infected cows. The number of BVDV infections in pigs on Jeju Island has decreased significantly since 2013 due to greater awareness in pig farmers and effective management of BVDV infection sources. In the Netherlands in the late 1980s, analysis of 700 samples from a slaughterhouse revealed that the seroprevalence of BVDV-strain Oregon was 20% [10]. A total of 10 years later, testing of 12,000 sows for the Dutch swine vesicular disease (SVD) surveillance program between 1993 and 2004 revealed a BDV/BVDV antibody seroprevalence of 11% [25]. Furthermore, samples from 6020 sows revealed a seroprevalence of 2.5% at the animal level and 11.0% at the herd level [11]. Changes in Dutch national policy which discouraged mixed breeding patterns blocked direct transmission of BVDV from cows to pigs, thereby reducing the prevalence of BVDV antibody-positive pigs [11].

In China, only three complete genomic sequences of BVDV strains isolated from pigs are available (SH-28, ZM-95, and SD0806). ZM- 95 [26] and SD0806 [27] are both BVDV-1 strains, while SH-28 is a BVDV-2 strain [28] and appears to be most similar to the Chinese cattle strain XJ-04 [29]. Although BVDV could not be isolated directly from pigs, in this study the pig infection experiment using BVDV-1a and -2a strains derived from cows was important as it confirmed that BVDV infection of pigs could occur indirectly. Therefore, additional surveillance and sequencing of BVDV strains in pig herds is necessary to determine the genetic relationship between strains derived from cattle and pigs, and their capacity for cross-species infection.

The signs and symptoms associated with BVDV infection in pigs are most often subclinical; the only evidence of infection is development of neutralizing antibodies to BVDV [9,18,30]. Experimental inoculation of animals with BVDV-1 and BVDV-2 demonstrated seroconversion and viraemia in pregnant gilts, but did not induce transplacental infection [31,32]. Other studies report that BVDV induces viremia at 7 days postinfection, with seroconversion 3 weeks after experimental inoculation [33,34,35]. The course of BVDV infection in pigs will depend on the virulence of the viral strain and the pig immune response [36], which may control the disease [37]. In this study, Jeju pigs infected with BVDV-1a or BVDV-2a exhibited no pathological features such as transient leukopenia, clinical signs, or organ/tissue lesions. In addition, the failure to detect BVDV in fecal and nasal samples infers that BVDV transmission is unlikely to occur between pigs. However, other papers suggest that the presence of the virus in the nasal secretions of infected pigs could act as a source of infection, thereby facilitating spread within the herd [32,38]. Although BVDV does not pose the same threat to pig herds as it does to ruminants, it may interfere with CSF monitoring and surveillance programs, leading to misleading diagnosis of the disease [11].

## 5. Conclusions

Here, we show that the route of BVDV infection among pigs is transmission from co-habiting or neighboring cows. When pigs were experimentally infected with cow-derived BVDV-1a or -2a strains, we detected BVDV antigens in several organs in BVDV-2a-inoculated pigs. These studies confirm that BVDV is circulating on pig farms and must be considered in Pestivirus control programs conducted in Jeju, South Korea.

## Figures and Tables

**Figure 1 vetsci-09-00146-f001:**
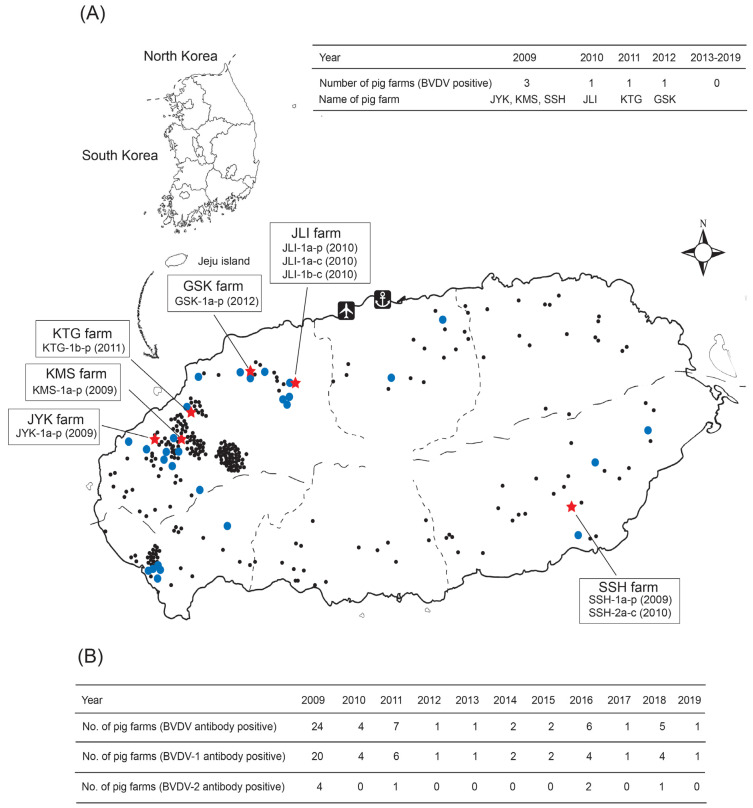
Bovine viral diarrhea virus–positive and BVDV antibody-positive pig farms on Jeju Island, 2009–2019. The locations and three letter designations of BVDV–positive pig farms are shown on the map (**A**). Number of BVDV antibody–positive pig farms (**B**). Pig farms, cow farms, and BVDV–positive pig farms are indicated by black dots, blue dots, and red stars, respectively.

**Figure 2 vetsci-09-00146-f002:**
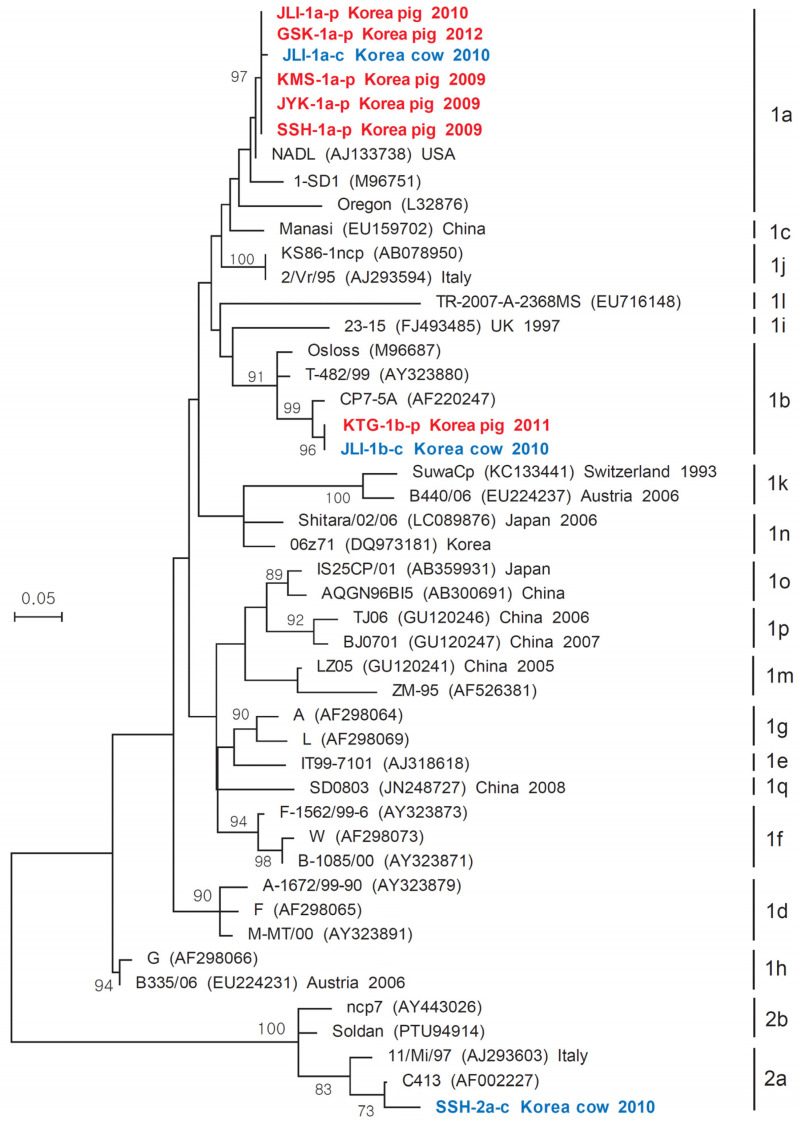
Maximum-likelihood tree analysis based on partial 5′UTR nucleotide sequences of BVDV strains isolated from pigs and cows. The partial 5‘UTR sequences of 9 BVDV strains isolated from Jeju Island pigs and cows in this study were aligned with 37 reference sequences from several different countries. BVDV strains isolated in this study are denoted by bold red and blue letters. The maximum-likelihood method (Tamura-Nei model) was constructed with 1000 bootstrap analyses in the MEGA 7.0 program.

**Figure 3 vetsci-09-00146-f003:**
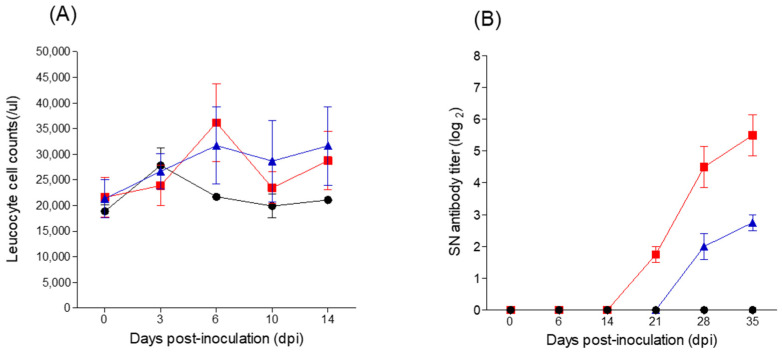
Changes in leukocyte cell counts and serum-neutralizing antibody titers in pigs experimentally infected with BVDV-1a and -2a. Leukocyte cell counts (**A**) and neutralizing antibody titers (**B**). Pigs inoculated with BVDV-1a, BVDV-2a, or no virus (mock infection) are shown as red squares, blue triangles, and black circles, respectively.

**Table 1 vetsci-09-00146-t001:** Detection of bovine viral diarrhea virus in fecal, nasal, and blood samples from pigs.

Group	Virus	No. of Pigs	Days Post-Inoculation (dpi)							
			0	3	6	10	14	21	28	35
1	BVDV-1a ^a^	4	-/-/-	-/-/-	-/-/1	-/-/-	-/-/-	-/-/-	-/-/2	-/-/1
2	BVDV-2a ^b^	4	-/-/-	-/-/-	-/-/-	-/-/-	-/-/-	-/-/-	-/-/1	-/-/-
3	Mock ^c^	4	- ^d^/- ^e^/- ^f^	-/-/-	-/-/-	-/-/-	-/-/-	-/-/-	-/-/-	-/-/-

^a^ BVDV-1a: bovine viral diarrhea virus type-1 (JLI-1a-c). ^b^ BVDV-2a: bovine viral diarrhea virus type-2 (SSH-2a-c). ^c^ Mock: no vaccine. Positive number of pigs from fecal ^d^, nasal ^e^, and blood ^f^ samples. -: negative.

**Table 2 vetsci-09-00146-t002:** Detection of bovine viral diarrhea virus antigens in pig organs and tissues by RT-PCR.

Group	Virus	No. of Pigs	To ^a^	Lu	He	Sp	Li	Ki	MLN	ILN	IL
1	BVDV-1a ^b^	4	-	-	-	-	-	-	-	-	-
2	BVDV-2a ^c^	4	-	1/4	-	1/4	1/4	2/4	1/4	1/4	1/4
3	Mock ^d^	4	-	-	-	-	-	-	-	-	-

^a^ To—tonsil; Lu—lung; He—heart; Sp—spleen; Li—liver; Ki—kidney; MLN—mesenteric lymph node; ILN—inguinal lymph node; IL—ileum. ^b^ BVDV-1a: bovine viral diarrhea virus type-1 (JLI-1a-c). ^c^ BVDV-2a: bovine viral diarrhea virus type-2 (SSH-2a-c). ^d^ Mock: no vaccine. -: negative.

## Data Availability

The gene sequences of the nine BVDV strains (Accession number: OM337786-OM337794) detected in pigs and cows on Jeju Island have been deposited in GenBank.

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
