# Peer review of "Prevalence of Bovine Viral Diarrhea Virus Infections in Pigs on Jeju Island, South Korea, from 2009–2019 and Experimental Infection of Pigs with BVDV Strains Isolated from Cattle"

_vetsci, 2022, doi:10.3390/vetsci9030146_

Round 1

Reviewer 1 Report

This paper is very interesting in that the authors' team successfully analyzed BVDV isolated from pigs in Jeju Island from 2009 to 2019 and learned that the antigen of this virus is similar to that of CSFV and is similar to BVDV-17 isolated from cattle in neighboring farms. This study is important for surveillance of BVDV infection in pigs in areas without CSF vaccine protection (e.g., Jeju Island), which will be very helpful for surveillance of pigs infected with viruses caused by these BVDV infections rather than CFSV, and will be very good for subsequent farm control.

I have some comments as the following, 

  1. What is the cause of BVDV infection in pigs in areas without CSF vaccine protection (e.g. Jeju Island)? As the discussion suggests that the infection may be transmitted by cattle infected with BVDV, is the route of transmission fecal-oral or droplet transmission?
  2. I would like to know the nucleotide sequences of BVDV-1a (SSH-1a-p, JYK-1a-p, KMS-1a-p and GSK-1a-p strains) isolated from pigs and JLI-1a-c and NADL isolated from cattle in Jeju Island as described in the results of 3.1, and whether there is any data to show how to determine the difference between anti-BVDV-1a and anti-BVDV-2a (lines 134-135 of the results in 3.1)
  3. the authors discussed that the antigen of BVDV is similar to that of CSFV, and the virus isolated from pigs in areas without CSF vaccine protection (e.g. Jeju Island) is BVDV but not CSFV, can the authors explain whether it is because the antibodies produced by pigs infected with BVDV can suppress CSFV infection. Can the authors explain this?
  4. Can Figure 3 indicate which curve is BVDV-1a, BVDV-2a, or no virus (mock infection), so that the reader can clearly know the result of Figure 3?

Author Response

Reviewer 1

This paper is very interesting in that the authors' team successfully analyzed BVDV isolated from pigs in Jeju Island from 2009 to 2019 and learned that the antigen of this virus is similar to that of CSFV and is similar to BVDV-17 isolated from cattle in neighboring farms. This study is important for surveillance of BVDV infection in pigs in areas without CSF vaccine protection (e.g., Jeju Island), which will be very helpful for surveillance of pigs infected with viruses caused by these BVDV infections rather than CFSV, and will be very good for subsequent farm control.

  1. What is the cause of BVDV infection in pigs in areas without CSF vaccine protection (e.g. Jeju Island)? As the discussion suggests that the infection may be transmitted by cattle infected with BVDV, is the route of transmission fecal-oral or droplet transmission?

Answer 1: BVDV transmission from infected cattle to pigs is presumed to occur. On Jeju Island, pigs and cattle are bred on the same farm, or pig farms and cattle farms are adjacent. Therefore, if a cow infected with BVDV excretes the virus in saliva and feces, it may infect pigs via movement of people, vehicles, and equipment.

  1. I would like to know the nucleotide sequences of BVDV-1a (SSH-1a-p, JYK-1a-p, KMS-1a-p and GSK-1a-p strains) isolated from pigs and JLI-1a-c and NADL isolated from cattle in Jeju Island as described in the results of 3.1, and whether there is any data to show how to determine the difference between anti-BVDV-1a and anti-BVDV-2a (lines 134-135 of the results in 3.1)

Answer 2: We revised the Results 3.1. “Anti-BVDV-1a and anti-BVDV-2a were tested the neutralization antibody using BVDV type 1a strain (08GB44-1, Accession no: JQ418633) or a BVDV type 2a strain (08Q723, Accession no: JQ418635) in Material and methods 2.4 (revised manuscript line 106-109). The two strains isolated from cattle in 2008 were used as BVDV antigens in the neutralization antibody test. We have attached the nucleotide sequences of the BVDV-1a (JYK-1a-p, KMS-1a-p, SSH-1a-p, JLI-1a-p, and GSK-1a-p) strains and the JLI-1a-c cow and NADL strains.

Data availability statement: The gene sequences of the nine BVDV strains (Accession number: OM337786-OM337794) detected in pigs and cows on Jeju Island have been deposited in GenBank.

>NADL 1a

gcaacagt ggtgagttcg ttggatggct taagccctga gtacagggta gtcgtcagtg gttcgacgcc ttggaataaa ggtctcgaga tgccacgtgg acgagggcat gcccaaagca catcttaacc tgagcggggg tcgcccaggt aaaagcagtt ttaaccgact gttacgaata cagcctgata gggtgctgca gaggcccact gt

>GSK-1a-p

gcagcagtggtgagttcgttggatggcttaagccctgagtacagggtagtcgtcagtggttcgacgccttggaataaaggtctcgagatgccacgtggacgagggcatgcccaaagcacatcttaacctgagcgggggtcgcccaggtaaaagcagttctaaccgactgttacgaatacagcctgatagggtgctgcagaggcccactgt

>SSH-1a-p

gcagcagtggtgagttcgttggatggcttaagccctgagtacagggtagtcgtcagtggttcgacgccttggaataaaggtctcgagatgccacgtggacgagggcatgcccaaagcacatcttaacctgagcgggggtcgcccaggtaaaagcagttctaaccgactgttacgaatacagcctgatagggtgctgcagaggcccactgt

>JYK-1a-p

gcagcagtggtgagttcgttggatggcttaagccctgagtacagggtagtcgtcagtggttcgacgccttggaataaaggtctcgagatgccacgtggacgagggcatgcccaaagcacatcttaacctgagcgggggtcgcccaggtaaaagcagttctaaccgactgttacgaatacagcctgatagggtgctgcagaggcccactgt

>KMS-1a-p

gcagcagtggtgagttcgttggatggcttaagccctgagtacagggtagtcgtcagtggttcgacgccttggaataaaggtctcgagatgccacgtggacgagggcatgcccaaagcacatcttaacctgagcgggggtcgcccaggtaaaagcagttctaaccgactgttacgaatacagcctgatagggtgctgcagaggcccactgt

>JLI-1a-p

gcagcagtggtgagttcgttggatggcttaagccctgagtacagggtagtcgtcagtggttcgacgccttggaataaaggtctcgagatgccacgtggacgagggcatgcccaaagcacatcttaacctgagcgggggtcgcccaggtaaaagcagttctaaccgactgttacgaatacagcctgatagggtgctgcagaggcccactgt

>KTG-1b-p

gcaacagtggtgagttcgttggatggctgaagccctgagtacagggtagtcgtcagtggttcgacgctttggaggacgagcctcgagatgccacgtggacgagggcatgcccacagcacatcttagcctggacgggggtcgttcaggtgaaaacggtttaaccaaccgctacgaatacagtctgataggatgctgcagaggcccactGT

>JLI-1b-c

gcaacagtggtgagttcgttggatggctgaagccctgagtacagggtagtcgtcagtggttcgacgctttggaggacgagcctcgagatgccacgtggacgagggcatgcccacagcacatcttagcctggacgggggtcgttcaggtgaaaacggtttaaccaaccgctacgaatacagtctgataggatgctgcagaggcccactGT

>JLI-1a-c

gcagcagtggtgagttcgttggatggcttaagccctgagtacagggtagtcgccagtggttcgacgccttggaataaaggtctcgagatgccacgtggacgagggcatgcccaaagcacatcttaacctgagcgggggtcgcccaggtaaaagcagttctaaccgactgttacgaatacagcctgatagggtgctgcagaggcccactgt

>SSH-2a-c

gcggtagcagtgagttcattggatggccgaacccctgagtacaggggagtcgtcaatggttcgacactcctttagtcgaggagtctcgagatgccatgtggacgagggcatgcccacggcacatcttaacccacgcgggggttgcatgggtgaaagcgccattcgtggcgtcatggacacagcctgatagggtgtagcagagaccctgtat

  1. the authors discussed that the antigen of BVDV is similar to that of CSFV, and the virus isolated from pigs in areas without CSF vaccine protection (e.g. Jeju Island) is BVDV but not CSFV, can the authors explain whether it is because the antibodies produced by pigs infected with BVDV can suppress CSFV infection. Can the authors explain this?

Answer 3: In general, antibodies specific for CSF and BVDV show cross-reactivity and trigger similar defense capabilities. It is predicted that the BVDV antibody is unlikely to inhibit CSF infection on Jeju Island. This is because the number of BVDV-infected pigs on Jeju Island is relatively low. However, if BVDV infection becomes widespread, we expect it to contribute to protection against CSF.

  1. Can Figure 3 indicate which curve is BVDV-1a, BVDV-2a, or no virus (mock infection), so that the reader can clearly know the result of Figure 3?

Answer 4: To clearly distinguish between BVDV-1a, -2a, and no virus (mock infection), we have revised the colors in Figure 3.

Reviewer 2 Report

Dear authors,

thank you very much for this interesting manuscript.

I would like to submit to your  attention some suggestions, as following.

  • LINE 53-55: in my opinion, it is necessary better explain the purposes of the study. Particularly, the correlation between the state of “CSF no vaccination area” and the proved “BVDV circulation in pig population” on Jeju Island is not very clear (also in the “conclusion” paragraph). Therefore, maybe it should be better explained the purpose of experimental infection.

  • LINE 57: in the title of paragraph 2.1, “blood” is missing.

  • LINE 58-60: please, explain if CSF serological positive samples have been tested for BVD, or if the 734 pig samples have been collected randomly.

  • LINE 88: please, explain if the 35 references are partial or complete, and also where they come from.

  • PARAGRAPH 2.5.: at line 112, different samples are reported; later, diagnostic use of such samples (feces and nasal swab) is not described. Therefore, could you explain why two inoculation vias (intranasally and intramuscularly) have been used?

  • FIGURE 1a: in the table, SSH farm is reported only in 2009, whereas in the map there is also SSH-2a-c in 2010.

  • FIGURE 1b: specification of farms where anti-BVDV1 antibodies and anti-BVDV2 antibodies were detected, is missing (as reported instead at lines 132-134).

  • LINE 143: “from around the world” should be included into “materials and methods” section, in 2.3. paragraph.

  • LINE 170: could you explain how BVDV was detected? PCR or isolation?

  • LINE 173: “fecal or nasal samples” should be reported also in “materials and methods” section.

  • LINE 215-218: please, report how long seroprevalence reduction programme lasted.

  • LINE 223: “BVDV1a and BVDV2a strains derived from cows” should be reported also in “materials and methods” section.

Author Response

Reviewer 2

Dear authors,

thank you very much for this interesting manuscript.

I would like to submit to your attention some suggestions, as following.

  1. LINE 53-55: in my opinion, it is necessary better explain the purposes of the study. Particularly, the correlation between the state of “CSF no vaccination area” and the proved “BVDV circulation in pig population” on Jeju Island is not very clear (also in the “conclusion” paragraph). Therefore, maybe it should be better explained the purpose of experimental infection.

Answer 1: Thank you. We have changed the title to “Prevalence of bovine viral diarrhea virus infections in pigs on Jeju Island, South Korea, from 2009–2019 and experimental infection of pigs for BVDV strains isolated from cattle,” as well as the study description on lines 53–56:

“The purpose of this study was to investigate the status and cause of BVDV infection in pigs in the Jeju island region from 2009 to 2019, and to help information BVDV infection by clinical observation, immunological and pathological analyzing experimental BVDV-1a and -2a infection patterns in pigs” (revised manuscript lines 57–60).

We have also revised the conclusion (lines 266–270).

  1. LINE 57: in the title of paragraph 2.1, “blood” is missing.

Answer 2: We changed the title of section 2.1: “Virus isolation from fecal samples” → “Virus isolation from samples.” We tried to isolate BVDV from blood samples from pigs and from fecal samples from cows.

  1. LINE 58-60: please, explain if CSF serological positive samples have been tested for BVD, or if the 734 pig samples have been collected randomly.

Answer 3: We have added the following text:

“There were a total of 168 pig farms with CSF antibody and antigen detection between 2009 and 2019. To identity BVDV-infected farms for CSF-antibodies positive pig farms, 734 (CSF-antibodies positive) pig blood samples were tested for BVDV antigen and antibody” (lines 66–69).

  1. LINE 88: please, explain if the 35 references are partial or complete, and also where they come from.

Answer 4: We added “The partial 5’UTR ~” (line 96).

  1. PARAGRAPH 2.5.: at line 112, different samples are reported; later, diagnostic use of such samples (feces and nasal swab) is not described. Therefore, could you explain why two inoculation vias (intranasally and intramuscularly) have been used?

Answer 5: The fecal and nasal samples were tested by RT-PCR. In addition, to ensure BVDV infection, each pig was inoculated with 1 ml via the intramuscular route and 1 ml via the intranasal route. We have revised the text as follows:

“Blood, feces, and nasal swab samples were collected from each pig at 0, 3, 6, 10, 14, 21, 28, and 35 days post-inoculation (dpi) and BVDV was detected by RT-PCR” (lines 126–128).

  1. FIGURE 1a: in the table, SSH farm is reported only in 2009, whereas in the map there is also SSH-2a-c in 2010.

Answer 6: The table in Figure 1a shows BVDV was detected only in pigs from the SSH farm in 2009. However, the map in Figure 1a indicates that BVDV was detected in cattle on the SSH farm in 2010.

  1. FIGURE 1b: specification of farms where anti-BVDV1 antibodies and anti-BVDV2 antibodies were detected, is missing (as reported instead at lines 132-134).

Answer 7: The number of pig farms with cases positive for anti-BVDV 1a and anti-BVDV 2a according to year has been included in the revised Figure 1b.

  1. LINE 143: “from around the world” should be included into “materials and methods” section, in 2.3. paragraph.

Answer 8: We have included this in the Materials and methods (2.3 paragraph) (lines 96–98).

  1. LINE 170: could you explain how BVDV was detected? PCR or isolation?

Answer 9: We have added “by RT-PCR” to the sentence (line 190).

  1. LINE 173: “fecal or nasal samples” should be reported also in “materials and methods” section.

Answer 10: We have included this sentence in the Materials and methods (in 2.5 paragraph) (lines 126–128).

  1. LINE 215-218: please, report how long seroprevalence reduction programme lasted.

Answer 11: We revised the sentences as follows:

“In the Netherlands in the late 1980s, analysis of 700 samples from a slaughterhouse revealed that the seroprevalence of BVDV-strain Oregon was 20% [10]. Ten years later, testing of 12,000 sows for the Dutch swine vesicular disease (SVD) surveillance program between 1993 and 2004 revealed a BDV/BVDV antibody seroprevalence of 11% [25]. Furthermore, samples from 6,020 sows revealed a seroprevalence of 2.5% at the animal level and 11.0% at the herd level [11]. Changes in Dutch national policy which discouraged mixed breeding patterns blocked direct transmission of BVDV from cows to pigs, thereby reducing the prevalence of BVDV antibody-positive pigs [11].” (lines 231–239).

  1. LINE 223: “BVDV1a and BVDV2a strains derived from cows” should be reported also in “materials and methods” section.

Answer 12: We added the following sentence to the Materials and methods (in 2.5 paragraph):

“The BVDV-1a and BVDV-2a strains used were isolated from cows” (lines 124).

Reviewer 3 Report

All sugestiom are in att.

IMPORTANT - the authors not inform about approval Ethical Commision to the experiment with use live animals - pigs.

Author Response

Reviewer 3

Review the manuscript Bovine viral diarrhea virus infections in pigs on Jeju Island, South Korea, from 2009–2019. 

  1. The authors present in the manuscript information about isolation BVDV from pigs on Jeju island and results the experimental infection pigs BVDV strain isolated form cattle farms on Jeju island. In opinion the reviewer the title should be correct.

Answer 1: We have revised the title as follows:

“Prevalence of bovine viral diarrhea virus infections in pigs on Jeju Island, South Korea, from 2009-2019 and experimental infection of pigs for BVDV strains isolated from cattle”.

Abstract

  1. line 15 – important is not CSF but BVD in pigs.

Answer 2: We have removed “which is antigenically related to classical swine fever virus (CSFV)” (= lines 15–16).

  1. line 24 – important in CFV serological monitoring in pigs is cross reaction with BVDV

Answer 3: We have revised the sentence as follows:

“ While BVDV infection is not particularly pathogenic in pigs, it is still important to monitor porcine BVDV infections due to a differential diagnosis of CSFV.” (lines 24–26).

Introduction

  1. line 30-37 – authors must be explained information about novel pestivirus species

Answer 4: We have added the following sentences:

“ Bovine viral diarrhea virus (BVDV) is a single-strand, positive-sense RNA virus belonging to the genus Pestivirus within the family Flaviviridae [1]. The genus Pestivirus includes animal pathogens that are of worldwide socioeconomic significance; these include BVDV (Pestivirus A–B), classical swine fever virus (CSFV, Pestivirus C), and border disease virus (BDV, Pestivirus D) [1]. Other Pestiviruses include Pestivirus E (pronghorn pestivirus), Pestivirus F (Bungowannah virus), Pestivirus G (giraffe Pestivirus), Pestivirus H (Hobi-like pestivirus), Pestivirus I (Aydin-like pestivirus), and Pestivirus J (rat pestivirus) [1].” (lines 29–37).

  1. line 49-52 – it’s not important, because in Jeju island use in CSF control serological monitoring

Answer 5: We have revised the text as follows:

“On Jeju Island, which is located off the southernmost tip of mainland South Korea, CSF vaccination has not been implemented since 1999. However, frequent detection of CSF-antibody-positive pigs was confirmed as being due to contamination of the live attenuated CSF vaccine strain [15]. However, some CSF antibody-positive cases are thought to be due to infection by BVDV, although this has not been reported formally”. (lines 51–56).

  1. line 53-55 – the aim of the study was experimental infection BVDV strains isolated in last year in Jeju island and explain clinical picture and immune response. You must change the aim of the study/manuscript.

Answer 6: We have revised the text as follows:

“The purpose of this study was to investigate the prevalence and cause of BVDV infection in pigs in the Jeju Island region from 2009 to 2019, and to provide information about BVDV infection via clinical observations and immunological and pathological analyses of BVDV-1a and -2a infection patterns in experimentally infected pigs”. (lines 57–60).

Materials and Methods 

  1. important is samples to BVDV isolated. About CSF only one line and in last part his point. If Jeju island are free CSFV you control animals or government monitoring this sitation ?

Answer 7: Jeju Island bans the introduction of pigs from other countries, including mainland in Korea, due to lack of CSF vaccination. Also, all pig farms (n = 300) on Jeju Island are tested for the presence of CSF antigens and antibodies 2–3 times a year. If CSF is detected, BVDV antigen and antibody tests are performed to confirm whether the results are caused by BVDV.

  1. How to make the prevalence. How may farm are in Jeju island? How many pigs live on Jeju Island? Only 734 pigs were control to BVDV antibodies? You isolated BVDV from 734 pigs during 10 years? line 62-63 - A total of 60 cow fecal samples (five samples per 62 farms from 12 cow farms) were also collected from cow farms in the vicinity of pig farms 63 with BVDV-infected pigs. While only 60 samples from cow were taken?

Answer 8: There are about 300 pig farms on Jeju Island, which have been monitored for CSF antigens and antibodies every year from 2009 to 2019. Over the 11 years, 734 CSF antibody-positive pigs have been detected on 168 pig farms. Of these, 171 were positive for anti-BVDV antibodies (n = 165) and BVDV antigens (n = 6). On 54 of the 168 pig farms (CSF antibody- and antigen-positive), BVDV infection was confirmed, with the number of positive cases ranging from one to six.

The reason that we collected only 60 samples from 12 cow farms is that we only collected diarrhea samples. 

  1. line 89 - if BVDV form GeneBank, that how may be BVDV1 and BVDV2

Answer 9: We have revised the sentence as follows:

“The partial 5’UTR nucleotide sequences of 37 reference BVDV from around the world generated groups 1 (subgroups 1a-1q) and 2 (subgroups 2a-2b) were obtained from GenBank” (lines 96-98).

  1. line 90 – BVDV reference strains – explain

Answer 10: We have added the following explanation:

“The partial 5’UTR nucleotide sequences of 37 reference BVDV from around the world generated groups 1 (subgroups 1a-1q) and 2 (subgroups 2a-2b) were obtained from GenBank” (lines 96–98).

  1. line 106 – information about microscope

Answer 11: We added the name of the instrument “Nikon Eclipse Ti (Nikon Instruments, Melville, NY)” to the revised manuscript (line 116-117).

  1. line 107-117 – if You prepared experimental design, you must explain origin of the pigs, how did you protect pigs against another infection, how were the pigs tested for other diseases before experiment and during experiment? or the pigs were SPF?

Answer 12: We have revised the text as follows:

“Healthy 70-day-old Jeju pigs free of porcine circovirus 2, porcine reproductive respective syndrome virus, CSFV and BVDV antigens and antibodies were used in this study. Animal experiment was performed at 25℃ temperature and 60-70 % humidity condition” (lines 119-121).

  1. More information about local place for pigs (humidity, temperature, ect.)

Answer 13: Please see the answer to the previous comment.

  1. Results
  2. line 119 – title will be information that it’s results BVDV in Jeju island.

Answer 14: We have revised the title as follows:

“…….antigens and antibodies against BVDV from pigs in Jeju Island” (line 133).

  1. line 194 – from you taken MLN: mesentery lymph node

Answer 15: We revised the text as follows:

“MLN: mesenteric lymph node; ILN: inguinal lymph node” (lines 206, 211–212).

  1. Discussion
  2. line 228-232 – You inform about results of Your experiment. In this part of the discussion, you will be confronted with results other authors, and there is no such thing.

Answer 16: We have revised the text as follows:

“The signs and symptoms associated with BVDV infection in pigs are most often subclinical; the only evidence of infection is development of neutralizing antibodies to BVDV [9, 30, 31]. Experimental inoculation of animals with BVDV-1 and BVDV-2 demonstrated seroconversion and viraemia in pregnant gilts, but did not induce transplacental infection [32, 33]. Other studies report that BVDV induces viremia at 7 days post-infection, with seroconversion 3 weeks after experimental inoculation [34-36]. The course of BVDV infection in pigs will depend on the virulence of the viral strain and the pig immune response [37], which may control the disease [38]. In this study, Jeju pigs infected with BVDV-1a or BVDV-2a exhibited no pathological features such as transient leukopenia, clinical signs, or organ/tissue lesions. In addition, the failure to detect BVDV in fecal and nasal samples infers that BVDV transmission is unlikely to occur between pigs. However, other papers suggest that the presence of the virus in the nasal secretions of infected pigs could act as a source of infection, thereby facilitating spread within the herd [33, 39]. Although BVDV does not pose the same threat to pig herds as it does to ruminants, it may interfere with CSF monitoring and surveillance programs, leading to misleading diagnosis of the disease [11]” (lines 249–264).

  1. Conclusions
  2. In this part the authors inform about the result you investigation – short and suitability the results to the future.

Answer 17: We have revised the text as follows:

“Here, we show that the route of BVDV infection among pigs is transmission from co-habiting or neighboring cows. When pigs were experimentally infected with cow-derived BVDV-1a or -2a strains, we detected BVDV antigens in several organs in BVDV-2a-inoculated pigs. These studies confirm that BVDV is circulating on pig farms and must be considered in Pestivirus control programs conducted in Jeju, South Korea” (lines 266–270).
